# Qualitative Assessment of Effective Gamification Design Processes Using Motivators to Identify Game Mechanics

**DOI:** 10.3390/s21072556

**Published:** 2021-04-06

**Authors:** Eva Villegas, David Fonseca, Enric Peña, Paula Bonet, Sara Fernández-Guinea

**Affiliations:** 1Group of Research on Technology Enhanced Learning (GRETEL), Universitat Ramon Llull, La Salle, 08022 Barcelona, Spain; enric.pc@salle.url.edu; 2Communities and Children Faculty, University of St. Mark and St. John, Plymouth PL6 8BH, UK; 20095694@marjon.ac.uk; 3Facultad de Psicología, Universidad Complutense de Madrid, 28223 Madrid, Spain; sguinea@psi.ucm.es

**Keywords:** effective gamification, engagement, motivation, design strategy, mechanics, gaming experience, user experience

## Abstract

This research focuses on the study and qualitative assessment of the relationships between motivators and game mechanics per the ratings of expert gamification consultants. By taking this approach, it is intended that during the design phase of a gamified system, decisions can be made about the design of the system based on the motivators of each of the profiles. These motivators can be determined from the information provided by the potential players themselves. The research presented starts from a previous analysis in which, based on the three most used gamification frameworks and through a card sorting technique that allows the user to organize and classify the content, a set of mechanics are determined. In the present study, each of the mechanics is analyzed, and a more precise motive is decided. As a result, a higher level of personalization is achieved and, consequently, approximates a higher level of gamification effectiveness. The main conclusions are implemented in the development of the Game4City 3.0 project, which addresses gamified and interactive strategies to visualize urban environments in 3D at an educational and social level.

## 1. Introduction

Systems based on user experience methodologies must consider the following factors: technological requirements; cognitive psychology in order to understand the user’s learning process of our design; and the design itself, which allows us to develop projects according to the necessary processes [1] because they are planned according to the motivators and needs of the potential users to whom they are directed [2]. The gamification of any content, system or environment requires prior studies to understand the technology and technique of the use of devices, sensors, systems or methods [3,4]. This overall statement is important because, depending on the dynamics to be gamified, the context or device to be used and finally the user’s profile, it is necessary to consider various variables that can influence the success or failure of a gamified proposal [4,5]. Starting from these premises, players and users become protagonists of the development process, promoting the creation of more effective systems from the beginning [6]. The research presented here is based on the study of the system’s future users. Therefore, in this case, the users become the gamification experts.

To analyze and establish the process, an iterative design system is used, which allows continuous and improved redesign from the previous design phase [7]. Therefore, we start by considering the discipline of user because the user’s experience will define the applicable methodology and technique; the expert thereby serves as the basis for the analysis to understand the mental model each expert adopts from the point of view of their work [8].

For the development of our objectives, we opted for a qualitative approach because we are initiating an exploratory process with the participation of sector consultants. This approach will be the most successful because it will allow us not only to identify and weigh data but also to explain their origins in depth [9,10,11].

The study is derived from an investigation of a methodology called I’M IN [12]. The methodology aims to indicate the process of designing a workshop session with users, incorporates the user experience as a behavioral evaluation, and includes the application of gamification as a method of improvement/enhancement of the motivations arising during the work session. The test that emerges raises the need to work with expert professionals in the sector when analyzing the necessary requirements, thereby generating data focused on the people who should be using them.

In parallel, the main conclusions of the project are implemented in the development of the Game4City 3.0 project. This project addresses gamified and interactive strategies for viewing urban environments in 3D at the educational and social levels [13]. The ability to create interactive and immersive virtual architectural models encourages a creative approach to architecture and fosters the acquisition of the skills needed to reproduce and understand the space [14]. Initially, some expert users in gamification who are related to the Game4City 3.0 project participated in our study as a first step towards understanding all motivational issues to leverage during the different phases of the project, which is designed to facilitate the participation of nonspecialized users and students in the visualization, understanding and reinterpretation of urban models and spaces, the promotion of user-centered design, and participation in design initiatives [15]. The main results and phases of the project have been widely published [16,17,18,19,20], which validated the motivational and user profile approach used. Adopting those previous results to the present work was justified because it was necessary to adapt the gamification as a function of the user and their needs [21].

According to Ryan, Rigby and Przybylski’s theory of self-determination in video games [22], gamification is used as part of the motivational design that can influence the behavior of the players based on the stimuli they receive; therefore, having useful information regarding motivators (which may include those of future users) can yield a very effective design proposal [23]. Innate psychological needs such as competence, autonomy and relationships, if satisfied, cause an improvement in self-motivation and mental health; conversely, when they are frustrated, they can produce a reduction in the motivation and well-being of a person [24]. Taking this parameter into account can provide added value for projects targeted for gamification. The term “degamification” was introduced by Margaret Robertson [25] who stated that in a gamified system, the commitment of the player to the game must be monitored. If the player is initially frustrated, they may stop participating in the challenge. Therefore, when the fictional elements of the games disappear, the game itself disappears as well. The decision in regard to choosing the mechanics, dynamics and game elements [26] is essential so that the player does not lose engagement [27] and stays immersed from the beginning to the end of the process. The various game elements can serve different motivational mechanisms, so several elements can be combined with these mechanisms [28]. For example, most games have a shelf life of 12 to 18 months, so in order to generate long-term commitment to a game, the key element of motivation is required.

On the same path and according to Mollick et al. [29], gamification separates the work that a person does from the work that they perceive themselves to be doing; thus, to understand gamification, it is necessary to understand the motivation for it. For this, the motive can be treated as a reward structure, and it can therefore define the behavior that the designer wants to encourage. It can help to train the person as a player—that is, to train the person to participate in the investment environment—and can help the player focus on tasks that are interesting to him.

The same concept is reaffirmed by Savignac [30], who indicated that gamification can improve the quality of life because it is a motivational tool and is therefore applicable to work environments.

This described link between gamification and motivation serves as a key feature in understanding the objective of the project (gamification key-points) that is to be proposed and the player profile for which it is designed because it reflects an emotional factor that drives a user to carry out an action.

Emotional behavior is based on the emotions that users experience while participating in the use of systems or projects, and these emotions are part of the human character. Using the definition of emotional behavior offered by Don Norman [31], three key factors of the users can be taken into account: the emotions evoked by the product during the interaction, the user’s mood and the user’s feelings associated with the product prior to the interaction.

In this sense, the user experiences various emotions during their interaction. The individual’s state of mind or humor can condition the emotions elicited by the product [32]. Taking into account the idea that user emotions can change over the course of their product interactions and be detected by a system, sensor, or device, our study is relevant because it provides gamification experts tools and allows their future improvement through the prism of knowledge surrounding the motivators related to user experience in gamified approaches. In this way, the article proposes new tools for experts by which to define the mechanics related to a gamified environment depending on the sensor or sensors used: fixed, mobile, multiuser environment, etc.

There are several studies that highlight the need to personalize gamified systems. Hexad, for example, is a system that allows a 24-question survey to define user preferences and that concludes in one of the following: philanthropist, socializer, disruptor, free spirit, completer or player [33].

According to Brave et al. [34], emotions are transmitted towards the use of the product during their interaction. Feelings are states not of the person but of the association that a person has with the product as a result of previous experiences with the same or similar products. Emotions affect attention and memorization capacity, user performance, and assessment. Therefore, the emotional aspects of the user influence the interactive use of the products from a hedonic perspective, as well as the cognitive processes [35]. The user experience as a methodology can be applied in several ways, and a clear example would be the evolution of virtual reality. The evaluation of the experience is carried out with various instruments: the Simulator Sickness Questionnaire (SSQ), the Fast Motion Sickness Score (FMS), the User Experience Questionnaire (UEQ-S) and the Virtual Reality Neuroscience Questionnaire (VRNQ) [36].

To address this problem, motivators are used as a conceptual framework. As presented in a previous study, the design of strategies based on motivators with a clear goal orientation results in an approach that encourages participation and the design of strategies [37]. From the approach of the design of strategies based on motivation, studies have focused on gamification wherein the main objective is based on enhancing motivation in the participation of users—this time, in learning environments [38]. The design of the technique is based on the link connecting gamification with motivation and commitment, and the results offer evidence on the effect of gamification on the motivation and commitment of students in three university courses [39].

When talking about the user’s previously associated feelings, one works from the point of view of intrinsic factors or motivational factors, that is, those feelings that are part of the individual, such as achievement, responsibility or recognition. These aspects can be conditioned by extrinsic factors, such as social and cultural elements, the product itself and the context of use. Therefore, the experiences that an individual has previously felt or lived can condition the feelings attributed to the product [40].

The link between user and player experience provides specific assessment tools in specific gaming environments, and there are several techniques that are recommended, particularly for games [41].

Therefore, gamification is linked to the will that drives effort with the interest in or purpose of achieving certain challenges. Because our behaviors are spurred mainly by motivations that give us the impetus to do something, deciding on the right game mechanics becomes a key element. A game must be voluntary, and motivation helps. As shown in Figure 1, as stated in engagement loops [42], when we work on motivation, action and feedback, feedback allows us to push back the motivation defined initially. According to the overlapping value structures, they determine that the type of player motivators must be known [43]; in this way, we avoid gamification remaining only on the surface.

The research starts from two initial premises: in light of the fact that there are several definitions of game mechanics and several gamification frameworks, can there be a consensus on game mechanics? In addition, game mechanics are directly related to provoking the motivations of the players. Can the most appropriate motivation be defined for each of the mechanics? The result of the study aims to establish a single criterion that allows applicability to various types of system [43].

## 2. Materials and Methods

### 2.1. Initial Concepts

The study is based on a study carried out as part of an investigation focused on providing gamification criteria to a methodology created for the evaluation of users in the field of user experience [44]. For example, game mechanics can be proposed as facilitators of the creation of calibration games holding the attentions of the users in a manner that is required in this type of task, thus maintaining the quality of the data [45]. Calibration kits are a novel way to motivate users to perform calibrations, thereby improving the performance and accuracy of many human-computer systems [46].

When designing a gamified product, the experts are the ones who choose the mechanics necessary to carry out the project according to the objectives and the profile of the players at which it is aimed, such that the consideration of the real motivations of each player provides very valuable information when choosing the item to include. Therefore, it considers two elements: game mechanics and motivators.

#### 2.1.1. Game Mechanics

We experimented with the hybrid card sorting technique, which focuses on diverging across data from several gamification frameworks that implement various game elements, in a phase prior to the one presented in this paper. We were able to extract a set of mechanics for categorizing, organizing, classifying, and hierarchizing content during this phase. The current research is based on the application of a technique that allows for the convergence of previous data, resulting in concrete results that enable successful design work. Figure 2 depicts the type of work done by two of the experts who were assessed in the previous phase [47].

The game mechanics used as a starting point are based on the results obtained in a hybrid card sorting analysis that starts as the basis for decisions about the meaning of the mechanics or the game elements, which are determined as the MDA system: mechanics, dynamics and aesthetics [26]. The purpose of the study is to identify the first impressions regarding the categorization of elements that are used to gamify and, therefore, represent the consensus drawn by experts in the sector. Three types of mechanics are considered [47]:Actionable Gamification Octalysis by Yu-Kai Chou [48]. Based on eight factors: meaning, empowerment, social influence, unpredictability, avoidance, scarcity, ownership, and accomplishment, with a total of 76 game mechanics. Unlike with the creation of the first version of cards, the total sub classifications of the eight indicated parameters were considered.Gamification Model Canvas by Sergio Jiménez [49]. Based on platforms, mechanics, dynamics, components, aesthetics, behaviors, players, simplicity, costs, revenues and a total of 59 cards distributed among the different mentioned parameters.Gamification Inspiration Cards by Andrej Marczewski [50], with a total of 54 cards, including options for mechanics and options for profiles.

From the 189 elements evaluated, the experts identified a total of 58 items to be selected as real mechanics. The mechanics obtained are shown below and are analyzed in this test. In Table 1, they are detailed in alphabetical order.

In the results of the study, the concept of storytelling is not considered game mechanics but is considered a key element as part of immersion in a gamification design.

#### 2.1.2. Motivators

In learning psychology, it is well known that one of the main factors that favors learning and information retention is precise motivation. This effect can be explained from the cognitive point of view and from the perspective of how the brain processes information. From the cognitive perspective, motivation favors the processes of attention and concentration that favor the acquisition of information, and the system is more receptive to store the information more adequately; therefore, more is retained. From the point of view of the brain, motivation activates the participation of brain structures, such as the limbic system and the hippocampus, which are closely related to the process of acquiring information, consolidating and storing it; therefore, it favors the player having better acting, learning and remembering [51,52,53].

In the case of the motivators used, because the decision of what motivates each user starts from a questionnaire previously made by the people themselves and therefore starts from rigorous and personalized information, the analysis of the motivational profile of Valderrama et al. is proposed [54].

The questionnaire is focused on human resources departments to determine the true motives or work behaviors of the workers. Even so, and as indicated by the same author, Beatriz Valderrama, the method can be adapted to research contexts [55]. The questionnaire is based on the model of the Wheel of motives by Beatriz Valderrama created in 2010, which shows five motives of approach and five of avoidance, which counteract the opposite motives, for a total of 10 dimensions [56].

Definition of approach motives:Autonomy: Preference for being independent, following one’s own criteria and making decisions for oneself.Power: Desire to lead others, compete and win, rise, receive admiration, have popularity and prestige.Achievement: Preference for overcoming challenges, achieving professional success and reaching high standards of excellence.Exploration: Interest in novelty and variety, seeking to learn and discover new ways of doing things.Contribution: Desire to help others, contribute to society and have a positive impact on the lives of others.

Definition of avoidance motives:Affiliation: Preference for being with others, being part of a group and feeling accepted.Cooperation: Desire to maintain egalitarian relationships, avoid inequity, and create distance from power, rivalry and abuse of power.Hedonism: Preference for saving effort and tension, avoiding sacrificing one’s own well-being to pursue goals.Security: Preference for maintaining the stability of the environment, as well as avoiding changes and uncertainty.Conservation: Desire to protect oneself, earn money and conserve material goods.

To understand the correspondence between approach grounds and avoidance reasons, Table 2 indicates the correspondence in a visual way. Table 2 shows the lists of motives according to the correspondence between them [56]:

### 2.2. Methodological Approach

The objective of the test is to obtain rigorous information thinking about the needs of the projects in which gamification is applied; for this, an approach in which the protagonist of the evaluation is the expert in this sector is chosen because such an expert has knowledge of the applicability of the system and understands the needs when choosing the best gamification properties, such as mechanics, dynamics and perception towards the players [56]. Therefore, a methodology of user-centered design (UCD) is applied [57].

The classic usability methods are based on the hypothetical-deductive paradigm, so the paradigm shift towards a less objective base perspective allows us to understand and explore the opinions of the participants evaluated during the test. This constructivist model adds value to the subjective model and permits the Socratic treatment of the individual. The user experience allows us to qualitatively assess the perspectives of the experts in the disciplines being studied, who truly are the future users of the study results [58]. A constructivist approach allows creating knowledge from the learning that emerged from the users themselves; under this premise, they create two tools: Create@School App (Clermont House, Wicklow County Campus, Rathnew, Ireland; available online https://www.createschool.ie/ (accessed on 1 March 2021)) and the Project Management Dashboard (PMD, available online http://webvantage.bvk.com/ (accessed on 1 March 2021)), which integrates game mechanics, dynamics and aesthetics in the study plan. The project measures satisfaction and usability based on user experience methodologies [59]. The techniques carried out from evaluation systems with users allow the project to evaluate a minimum of five users of the same profile, and an evaluation based on qualitative results allows results obtained from User 6 to be repeated with previous users [60]. The study that is presented has a rating of 14, which supports the rigor of the results and validates the results of the assessments.

#### 2.2.1. Definition of the Evaluated User Profile

Taking into account several existing works on sample size in qualitative studies, there are two outcomes. On the one hand, the technique applied for data collection and on the other hand, always taking into account the saturation principle, the analysis of these data [61]. This principle is based on defining to what extent the data collection is providing new information in terms of the setting and the possible sample. The aim of qualitative research is to understand people’s perceptions in depth and to understand a detailed description. Therefore, a larger sample size may lead to a repetition of information [62].

It is important to highlight that for the study, the decisions of the professionals are critical, so special attention has been given to ensuring that all participants are highly qualified, that they are actively part of the sector and that they have a certain prestige in the field in which they work. Moreover, it is emphasized that to motivate the completion of the questionnaire, the motivation of the users of their projects is clearly and uniformly valued.

Some studies, such as those that evaluate people with autism spectrum disorder, allow us to understand the importance of being specific in the profile to provide concrete solutions to specific needs [63,64].

Regarding the number of experts required, there is no unanimous agreement for their determination. In Cabero-Almenara et al. [65], the selection of the number of experts depends on aspects such as the ease of accessing them or the possibility of having enough experts on the subject under investigation. On the other hand, authors such as Escobar-Pérez et al. [66] point out that the number of judges to be employed in a trial depends on the level of expertise and the diversity of knowledge. For this, expert judgement is used, which is defined as an informed opinion of people with experience in the subject, who are recognized by others as qualified experts in it and who can provide information, evidence, judgements and evaluations [65]. The characteristics of the expert must be defined, and at the same time, the number of them must be determined. According to Delgado-Rico et al. there must be at least three [67].

The test is carried out with a homogeneous group of people, consisting of 14 expert consultants in gamification [60] with more than five years of experience in business projects, educational projects or research projects and with a standard deviation of 3.33 with respect to age. It is important that the study be based on such a specific profile that, as previously explained, would consist of the potential users of the study results.

#### 2.2.2. Consent of Respondents

Prior to the sessions, information was provided on the research project along with information about the questionnaire that they were going to be administered; signed consent was requested from all consultants. During all phases of the research, special care was taken with the treatment of data both at an ethical level and in the consent of all participants for each action of the research. To formally comply with these processes, the research was overseen by the ethics committee of Universitat Ramon Llull, both for the information detailing the research and the type of informed consent given to the participants. The data obtained from participation were not used for any purpose other than that specified in this research; they were used exclusively for academic purposes and were kept in absolute confidentiality.

#### 2.2.3. Applied Methodology

The applied methodology was designed specifically for the study; therefore, it was specifically fashioned in light of the needs to be reflected in the results. The methodology is based on the telematics dissemination of a questionnaire for each expert to complete individually. Apart from the support information, the questionnaire consists of two steps:Step 1: This step includes a list of 58 predefined and indicated game mechanics, as shown in Table 1. The order is impartial; therefore, the list is presented alphabetically. A checklist allows quick selection of each mechanic. Each field that is selected is reinforced with a label that highlights that it is a game mechanic. In this step, the consultants are encouraged to re-evaluate their decisions when considering the game mechanics so that the consultants themselves are sure of the work they are doing; this procedure ensures criteria that are rigorous and necessary for the second step.Step 2: In this step, respondents link each item selected as a mechanic in Step 1 with one of the 10 motives raised and indicated in Table 2. The proposed mechanics are automatically indicated from the sample selected in the previous step; in this way, it is guaranteed that the consultant works only from the selection made by himself during the previous step. Each of the included motives is defined by the author so that there are no doubts about the interpretations of any of the consultants. For the selected mechanics, the names may be changed if necessary, and the consultants are asked to define each mechanic to clarify the link indicated with the motive. Once defined, the consultant selects the most related motivator from a list field.

For the result to be consistent among the experts, the objective of the survey is initially explained, and the source of the game elements of the first step is described. Definitions for each of the motives are provided so that differing interpretations do not arise.

## 3. Results

### 3.1. Step 1. Selection of Game Mechanics

The results obtained in the first step are provided in detail in Table 3, which shows the percentage of consultants who selected each item. Sharing knowledge was the most valued item for game mechanics, with a 93% selection rate, followed by Exploration, Gifting/Sharing and Voting/Voice, at 64%—that is, selected by more than half of the consultants. Among the concepts, a broader degree of dispersion exists with regard to selections made by almost half of the consultants (i.e., from 43% to 21%): Boss fights, Competition, Milestone unlocks, Miniquests, To attend an appointment, To win a reward, Build from scratch, Challenges, Exchangeable points, Group quests, and To enhance identity. The items for which a smaller number of consultants were in concurrence (i.e., selected in 7% to 14% of evaluations) were Inventory, Leader board, On-boarding/Tutorials, Social discovery, Social network, Social status, Unlockable/Rare content, Access, Catalogue, Innovation platform, Meaning/Purpose, Signposting and Virtual goods.

According to the observed degree of agreement in the selection of mechanics, it can be seen that most consultants considered the elements in bold in Table 3 to be those that they would use as game mechanics, for a total of 15 elements.

In the same way, the elements that no consultant appreciated as a game mechanic are highlighted: Status points, Status quo sloth, Theme and Virtual currency. These elements are considered or appreciated as game elements but cannot be used as mechanics.

### 3.2. Step 2. Motivators According to Mechanics

From the list that the consultants considered mechanisms per Step 1, the links that each consultant identified with the most-related motivator are analyzed. These motivators are indicated in Table 2. A list of the motivators is provided in Table 4 along with the percentage of their selection (and, therefore, of their association), based on the mechanics. Achievement is the most selected motive at 55%, followed by much lower values for Autonomy and Hedonism at 28% and Affiliation and Power at 26%. The motivators least appreciated among the selections were those of Contribution at 22%, Cooperation and Exploration at 19% and, least linked, the motivators of Conservation at 14% and Certainty at 10%.

### 3.3. Motivators Linked to Mechanics

From the selection of the game mechanics, the percentage of agreement of the experts with respect to the presented list of motivators is indicated in Figure 3. In this way, the motivators most closely related to each of the mechanics are known.

The percentage of motivators linked to Sharing knowledge was highest for the Cooperative motivator (at 50% of the links), followed by 34% for Contribution and 8% for Power and Affiliation.Exploration had 75% of its linkages made to the Exploration motivator, with the 25% remaining linkages being made to Autonomy and Achievement.For the Gifting/Sharing mechanic, the Contribution motivator was noted by 50% of the consultants, whereas only one-quarter (25%) selected the Cooperation motivator, and even smaller percentages associated the mechanic with the motivators Hedonism and Achievement. In Figure 3, in the case of Voting/Voice the results were more even, with 40% being linked to Contribution and Power and 20% to Autonomy.The mechanic Boss fights was linked to Power by 50%, to Achievement by 38%, and, at a much lower level, to Affiliation by 12%, as seen in Figure 3.The Competition mechanic was linked to the power motivator by 67% and to Achievement and Contribution by far fewer (17% and 16%, respectively).The selection rates for the Milestone unlocks mechanic were 50% for Achievement, followed by 37% for Conservation and 13% for Contribution.For Miniquests, as indicated in Figure 3, four motivators were indicated: Exploration at 50%, Achievement at 25%, Certainty at 13% and Cooperation at 12%.To attend an appointment, per Figure 3, was dispersed to five different mechanics, but the most linked was Affiliation at 50%, with the remaining percentage distributed equally across Autonomy, Cooperation, Achievement and Exploration.For the evaluation of To win a reward, the selection of Achievement as a motive stood out at 56%, followed by lower percentages for Conservation (33%) and Power (11%).In the case of Build from scratch, there was much dispersion in the results because there was no clear agreement among the experts. Achievement’s value was the highest at 37%, followed by Autonomy at 25% and the motives of Certainty, Exploration and Contribution at 13%.The attribution of the mechanic Challenges was very clear, with 63% agreement on achievement. The rest of the percentage was distributed among three motivators: Affiliation, Cooperation and Power.In the case of Exchangeable points, the motivator that the consultants most frequently selected was cooperation at 57%. The rest of the motivators had lower percentages, between 14% and 15%: Power, Achievement and Contribution.Match rates of 38% and 37% were made for the motivators Hedonism and Autonomy, respectively, with the rest of the percentage distributed to Achievement (13%) and Affiliation (12%). The Group quest mechanic was linked 100% to the motivator Cooperation.

### 3.4. Mechanics and Motivators

From the indicated results, the selected percentages for each mechanic are shown, classified according to whether they are closeness motives or avoidance motives [55]. The experts expressed their rationales for why they selected the motivator according to the mechanics with which they were already familiar and that they had already used. Below are some exact sentences taken directly from the comments.

Competition was defined as a “Motivation tool” when it was linked to Achievement, as “Competing when overcoming a certain challenge”, “Challenge between two or more people in which only one can win”, “Compete to achieve a goal” and “Relating to the competition between the player vs system or player vs player” when linked to Power and as a “Collective challenge to solve a problem” when linked to Contribution.

To win a reward was defined as “Prizes through achievements”, “Get a reward for completing a challenge”, “Reward” and “Earning a reward after overcoming a challenge” when linked to Achievement and as “Having ability, an action that gives the ability to do something” when linked to Power.Exploration was defined as follows: “Give the user the opportunity to know the details of a system by himself" in the case of Autonomy; “Relative to the fact that the player explores the system”, “Exploring the environment in order to learn to solve a challenge”, “Explore a play space”, and “Short investigation that seeks the observation of a faction to explore a space, site, information seeking to discover something/someone" in the case of Exploration; and "Search, observe, analyze, synthesize, elaborate, decide and, ultimately, practice a good amount of skills thanks to a goal and a certain lack of means to get it” in the case of Achievement.Gifting was defined as “Gifts from a user to other users” in the case of Hedonism; “Sharing achievements or knowledge that allow overcoming new challenges”, “Prize” and “Action of exchanging objects between different players or characters of a team controlled by a single player” in the case of Cooperation; and “Relating to the action of sharing something with other players” in the case of Contribution.Boss fights were defined as “Relating to the action of fighting, in this particular case doing it against”, “Bigger than normal challenge, usually at the end of a stage”, “End of level challenge where a player can demonstrate all the skills learned and use all the elements obtained during the same and previous challenges”, "Although it can be against other types of enemies, even with other players” and “I understand that it refers to the mechanics of facing a challenge with a higher level of difficulty (difficulty concentration)” when related to Power.Sharing knowledge was defined as “Relating to the fact of sharing, but in this case when sharing information—for example, with other players, since they do not know it" and “Action of sharing knowledge that can be parameterized within the game or that cannot be parameterized, in which could constitute some kind of Milestone Unlock” when linked to Contribution, “Sharing what is known to achieve a greater good and thanks to the cooperation of an entire community” in the case of Cooperation, and “Many classic board games use this mechanic of evaluating knowledge. Dangerous by the way” in the case of Power.To attend an appointment was defined as “Relating to the action of attending, but in this case to attending an appointment; it can be understood as a game mechanic if it is based on attending appointments, for example” when it was related to Autonomy, “Reward yourself for being there and on time, to contribute and/or participate in a certain way in a certain activity. It can lead to the knowledge of new opportunities, people and/or content” when related to Affiliation, “Understand mechanics as behaviors and actions that the players must do. Therefore, a mechanic would fit here” when related to Achievement, and “The possibility of acting and thinking without depending on something or others” when related to Autonomy.To enhance the identity was defined as “Relating to the expansion/development of a specific skill, such as that which could occur, for example, in a skill tree” and “The possibility of acting and thinking without depending on something or others" when related to Autonomy, “Reward yourself for being there and on time, to contribute and/or participate in a certain way in a certain activity. It can lead to the knowledge of new opportunities, people and/or content” in the case of Affiliation, and “Understand mechanics as behaviors and actions that the players must do. Therefore, a mechanic would fit here” in the case of Achievement.Milestone Unlocks was defined as “I like to think of the mechanical part as the way that players interact with the game elements (and with the game itself). Therefore, an unlocking of an achievement (mechanic) fits to overcome a level (element)”, “Milestones to be achieved within a process”, “Point that unlocks new game elements”, “Action to remove impediments that hinder the progress of an action to obtain an expected result”, and “Challenges that unlock new challenges, new mechanics, new tools and/or exploration spaces” when linked to Achievement.Miniquests were defined as “Small challenges” and “Actions to remove impediments that hinder the progress of an action to obtain an expected result” when linked to Achievements and “Small noncompulsory alternative missions that can enrich the gameplay and storytelling experience while refreshing the player with more relaxed levels of difficulty” when linked to Security.Build from scratch was defined as “Construction from blocks, physical or conceptual, of one" and “Start from the beginning” when linked to Autonomy and “I guess he refers to when building from scratch in strategy or farming games" when linked to Conservation.Exchangeable points were defined as “Action of exchange of items between monetary goods in the game (rupees) and goods for sale (weapons, armor, potions, etc.)” in the case of Security.Group quests were defined as “Challenges to be solved in a group” and “Missions shared between different players in a multiplayer context” in the case of Affiliation.Challenges were defined as “End of mission, challenges, tests to overcome” when linked to Achievement, “Challenges to be overcome by the user” when linked to Power, “Challenges that test the player’s skill and motivate them to keep playing” when linked to Affiliation and “Introduce challenges to solve in a period of time” when linked to Cooperation.

Table 5 and Table 6 show the game mechanics defined by the consultants and their relationship with the motivators categorized by closeness and avoidance. As previously indicated, the selection of mechanics does reflect the consensus of all; therefore, in the sum of percentages, there may be mechanics that do not show 100% consensus across the selected motivators.

In Table 5, the mechanics related to Power stand out, such as Competition, with 67%, and Boss fights, with 50%. In the case of Achievement, Challenges were cited 63% of the time, To win a reward 56% of the time, Milestone unlocks 50% of the time and Build from scratch 37% of the time. The Exploration motive was primarily associated with the Exploration mechanic (75%) and Miniquests (50%). The Contribution motive was associated with the case of Gifting/Sharing 50% of the time. In the case of the Voting/Voice mechanic, 40% of associations were with Power and the same percentage with Contribution.

In the case of the avoidance motives, as indicated in Table 6, for the Sharing knowledge mechanic, Cooperation was proposed 50% of the time; for the For attending an appointment mechanic, Affiliation was proposed 50% of the time; for the Exchangeable points mechanic, Cooperation was proposed 57% of the time; for the Group quests mechanic, Cooperation was proposed 100% of the time; and for the To enhance identity mechanic, Autonomy and Hedonism were proposed equally often (37% and 38%, respectively).

### 3.5. Summary of the Links between Mechanics and Motives

In summary, Table 7 indicates the list of mechanics together with the motives linked at the highest relative frequencies.

Table 8 indicates the mechanics related to the different motivators.

## 4. Discussion

In this section, the results obtained from the study are analyzed, focusing both on the evolution of the selection of game mechanics and the study of the link with the most related motives.

The selection of game mechanics is based on a study carried out previously using the card sorting technique. This study is based on an aggregation of three of the most commonly used mechanisms in gamification projects: Octalysis de Yu-Kai Chou, Gamification Model Canvas by Sergio Jiménez and Gamification Inspiration Cards by Andrej Marczewiski. The results identified a total of 58 items considered to be game mechanics according to the experts; these serve as the starting point for the analysis presented. Working with experts in the project allows us to focus the results on an objective they might find useful and necessary for their new projects in the gamification sector.

As reflected in previous studies, gamification can be a key element to motivate or involve people to achieve the challenges that are proposed to them, for example, in software development organizations. It has a direct impact on productivity and the quality of production. The entire contribution of the person is based on intrinsic motivation [68].

The study starts from a review of the previous selection wherein experts are asked to evaluate the proposed items according to whether they consider them to be game mechanics. The result shows a great dispersion of opinions, so with a consensus of a minimum of 50% of the professionals, a total of 15 mechanics have been obtained, requiring a total of 43 proposed items from the previous test to be thrown out, out of which 17 items have a percentage of coincident selection of 14% or less.

From the selected mechanics, links with the proposed motives are indicated. The most selected motive, with a 55% link, is that of Achievement. According to the definition of gamification and the main objectives, this makes sense because a clear motivation is to overcome challenges and achieve success or excellence. Consequently, many designs are precisely based on this type of objective, and for this, different game elements are applied. The next most linked motives, at 28%, are Autonomy and Hedonism. On the one hand, Autonomy is based on an approach motive, and there is a certain preference for being independent and making decisions autonomously. In the case of Hedonism, which is based on the avoidance motive, the achievement of challenges is pursued, but with some ease, saved effort and reduced tension. In both cases, the personal valuation of the players is appreciated, and in a certain way, these two are also linked to the motive for overcoming challenges through making decisions. With lower percentages of selection are Affiliation, Power and Contribution; these are motivators whose focus lies in the preference to be with others and form a group, with the desire to lead others to win and succeed and with the desire to help and contribute. In this case, as in the case of Achievement, these are concepts closely linked to the objectives of why projects are gamified. The list continues with the values of Cooperation, Exploration and Conservation; in these cases, the motivations are focused on maintaining a type of relationship with others in an egalitarian way, with an interest in variety, innovation and learning and with the value of conserving one’s own property. The motive least linked to mechanics is Certainty; maintaining security and avoiding uncertainty is not a parameter very closely related to gamification, but even so, at 10%, it appears to be important for at least some mechanics.

Taking into account the qualitative evaluation—which captures the point of view behind the choice and offers definitions of the game mechanics—allows us to understand their indicated relations with the motivators. Therefore, our approach allows us to understand not only the result but also the motive for the result.

From these results, the percentages of motives selected according to the game mechanics are indicated. The significant value of this type of information is based on the fact that the selection of an appropriate element for a gamified design is decided based on the motives personalized according to the players’ profiles. Therefore, it is key information to be effective in the initial phase of a proposal.

The mechanism with the greatest consensus in the selection of the motive is that of Group quests, with a 100% percentage link to Cooperation; knowledge of this motive can add value in how to include this parameter. This motive is centered on the desire of users to maintain a type of egalitarian relationship, thereby avoiding power and rivalry.

The mechanics with the highest consensus are those of Exploration, Competition and Challenges. In the case of Exploration, the linkage is 75% with Exploration. Although this is a high percentage, not linking it with the eponymous motive would be striking. By definition, Exploration describes a user who likes to explore, learn and discover. However, for this mechanic, some of the consultant’s highlight autonomy when making decisions and the achievement or objective in achieving challenges. Competition is mostly linked to the desire to help others and contribute to society; therefore, experts appreciate competition in the field of gamification outside of power and achievement. In the case of Challenges, a concept focused on Achievement is appreciated.

The game mechanics with less consensus among experts is that of Build from scratch because it is divided among five different motives: Autonomy, Achievement, Certainty, Exploration and Contribution. All of these motives are based on the preference to be independent, to achieve challenges to achieve excellence, to achieve security with an interest in learning and the desire to help others.

As a result of the study, a classification of the mechanics is obtained with the priority motive. From these data, experts can choose the appropriate mechanics for their potential players and can, in this way, ensure more effective gamification.

## 5. Conclusions

Carrying out the study using the evaluations of gamification experts allows us to obtain subjective results through quantitative data and related qualitative data. Developing a study leveraging an iterative design and the participation of experts is key to adding value to this type of proposal and, therefore, was a decision needing to be made in the design phase.

The realization of the study of motivators using a questionnaire administered in the market of potential players allows us to achieve rigor in the results obtained and very valuable information when making decisions. When decisions are made with game mechanics, potential players have the necessary association to understand the profile.

It would be very beneficial to obtain a certain consensus in the materials that can be used at the design level of a gamified product to select the most used game mechanics. The research has made it possible to understand the need for a study that would allow the extraction of the items, but even so, a second study has again seen differing opinions among the experts.

The motivators should not be interpreted as enhancing elements alone; they can also be elements to mitigate. If the stimuli of the player profiles are known, the experts, depending on the objectives of the project, can decide what they are interested in using.

Engagement is essential when considering this type of system, so understanding the stimulus and real emotions of the players allows the creation of a more effective system. In the case of the application of the results in our current Game4City 3.0 project, we have taken into account the need to adapt the interaction of the gamified system as a function of the user to maximize the engagement. Additionally, we have identified how different types of profiles use a gamified environment and interact in significantly different ways, illuminating the need to personalize this interaction. Students perceive the use of new representative systems to have great exploitative potential for their architectural projects and have high motivation to continue their training in these methods and systems. However, once specific training has been implemented, such as that proposed in the subject, there is still a certain resistance to its widespread use, even when accompanied by high levels of satisfaction and aid in the spatial understanding of the project.

By analyzing engagement with teachers and professionals, it can be perceived that part of the problem lies in the lack of training of professors in these systems. Typical professors who use CAD and photographic composition systems do not have the necessary training in new technologies that allow their use to be transversally promoted within their subjects. Due to this lack of use, students do not perceive the usefulness of advanced tools within their training and sense them as complex training. The present study could help define a gamified system adapted to the motives of the students and thus propose a more effective type of education focused on their needs. On the other hand, the individuals who have played with the Game4City 3.0 proposals describe a high level of engagement, autonomy, exploration and valuable contribution to the space design, identifying different aspects that were not contemplated in the initial architectural and urban projects. These positive results can be related to the previous experience realized by the game designers who participated in the current experience, who have identified some motivators to empower in the design and implementation of the architectural project.

As a next step and as part of the iterative design process, the results obtained should be analyzed in the application of a project design to understand their effectiveness in decision making, with regard to the interpretation and training of experts before the evaluation of the results of the motivation questionnaires, and for the degree of effectiveness of the gamified design.

## Figures and Tables

**Figure 1 sensors-21-02556-f001:**
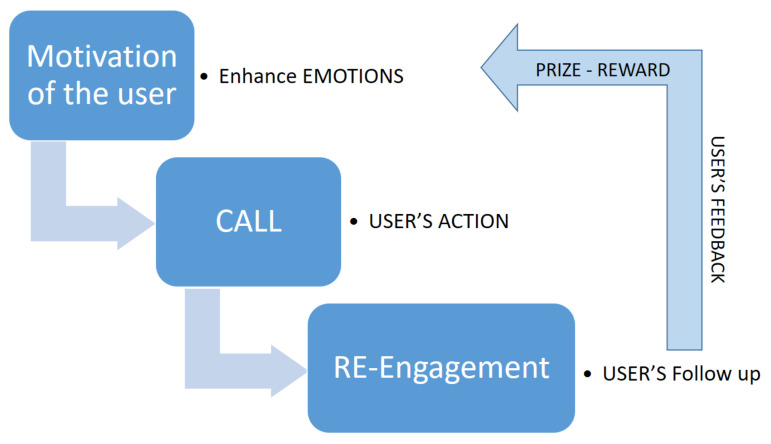
Representation of Core engagement loop model.

**Figure 2 sensors-21-02556-f002:**
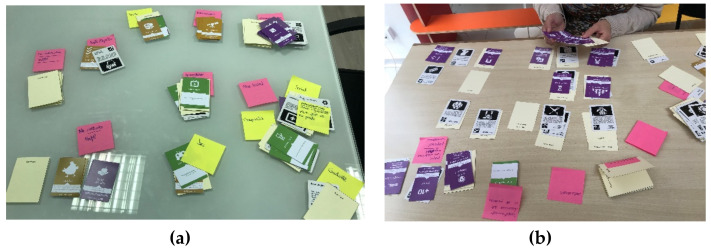
Previous evaluation with the Open Card Sorting Method: (**a**) image of the classification of mechanics made by User 2; (**b**) image of the classification of mechanics made by User 3.

**Figure 3 sensors-21-02556-f003:**
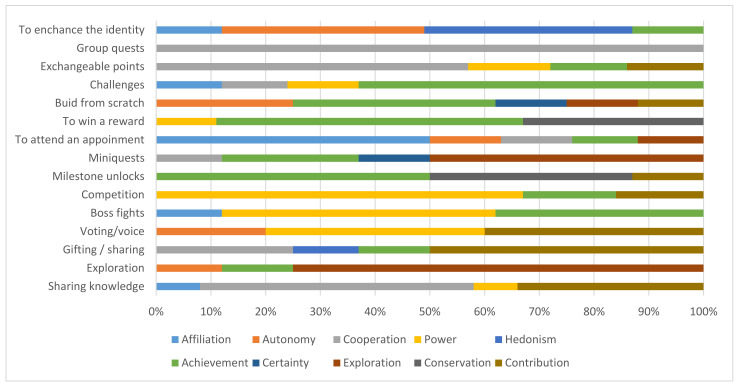
Percentage of motives related to mechanics.

**Table 1 sensors-21-02556-t001:** List of mechanics used in the test.

Game Mechanics			
1	Access	30	Meaning/Purpose
2	Achievements	31	Milestone unlocks
3	Avatar	32	Miniquests
4	Badges	33	On boarding/Tutorials
5	Boss fights	34	Points/Experience points (XP)
6	Branching choices	35	Progress bar
7	Build from scratch	36	Quest lists
8	Caretaking	37	Real prizes
9	Catalogue	38	Real-time control
10	Challenges	39	Sharing knowledge
11	Collection sets	40	Signposting
12	Competition	41	Social discovery
13	Count down	42	Social network
14	Creativity tools	43	Social status
15	Customization	44	Social treasure/Gifting
16	Development tools	45	Status points
17	Easter eggs	46	Status quo sloth
18	Exchangeable points	47	Step-by-step overlay tutorial
19	Exploration	48	Theme
20	Free lunch	49	Time dependent
21	Gifting/Sharing	50	To attend an appointment
22	Group quests	51	To enhance the identity
23	Guilds/Teams	52	To visualize the progress
24	High five	53	To win a reward
25	Innovation platform	54	Unlockable/Rare content
26	Inventory	55	Virtual currency
27	Leader board	56	Virtual goods
28	Levels/Progression	57	Virtual storytelling
29	Lottery/Game of chance	58	Voting/Voice

**Table 2 sensors-21-02556-t002:** Correspondence between closeness and avoidance motives.

Closeness	Avoidance
Autonomy	Affiliation
Power	Cooperation
Achievement	Hedonism
Exploration	Certainty
Contribution	Conservation

**Table 3 sensors-21-02556-t003:** Game mechanics selected by the experts. Bold marks mechanics with selection rate ≥50%.

Evaluation of Game Mechanics			
**Sharing knowledge**	**93%**	Quest lists	36%
**Exploration**	**64%**	Real-time control	36%
**Gifting/Sharing**	**64%**	Achievements	29%
**Voting/Voice**	**64%**	Creativity tools	29%
**Boss fights**	**57%**	Free lunch	29%
**Competition**	**57%**	Points/Experience points (XP)	29%
**Milestone unlocks**	**57%**	Social treasure/Gifting	29%
**Miniquests**	**57%**	Time dependent	29%
**To attend an appointment**	**57%**	Avatar	21%
**To win a reward**	**57%**	Badges	21%
**Build from scratch**	**50%**	Development tools	21%
**Challenges**	**50%**	Real prizes	21%
**Exchangeable points**	**50%**	Step-by-step overlay tutorial	21%
**Group quests**	**50%**	Virtual storytelling	21%
**To enhance the identity**	**50%**	Inventory	14%
Branching choices	43%	Leader board	14%
Caretaking	43%	On boarding/Tutorials	14%
Collection sets	43%	Social discovery	14%
Customization	43%	Social network	14%
Easter eggs	43%	Social status	14%
High five	43%	Unlockable/Rare content	14%
Levels/progression	43%	Access	7%
Lottery/game of chance	43%	Catalogue	7%
To visualize the progress	43%	Innovation platform	7%
Count down	36%	Meaning/Purpose	7%
Guilds/teams	36%	Signposting	7%
Progress bar	36%	Virtual goods	7%

**Table 4 sensors-21-02556-t004:** Motivators according to the percentage of links made with the game mechanics.

Motives	
Achievement	55%
Autonomy	28%
Hedonism	28%
Affiliation	26%
Power	26%
Contribution	22%
Cooperation	19%
Exploration	19%
Conservation	14%
Certainty	10%

**Table 5 sensors-21-02556-t005:** Motives of Closeness.

	Autonomy	Power	Achievement	Exploration	Contribution
Sharing knowledge		8%			34%
Exploration	12%		13%	75%	
Gifting/Sharing			13%		50%
Voting/Voice	20%	40%			40%
Boss fights		50%	38%		
Competition		67%	17%		16%
Milestone unlocks			50%		13%
Miniquests			25%	50%	
To attend an appointment	13%		12%	12%	
To win a reward		11%	56%		
Build from scratch	25%		37%	13%	12%
Challenges		13%	63%		
Exchangeable points		15%	14%		14%
Group quests					
To enhance the identity	37%		13%		

**Table 6 sensors-21-02556-t006:** Motives of Avoidance.

	Affiliation	Cooperation	Hedonism	Certainty	Conservation
Sharing knowledge	8%	50%			
Exploration					
Gifting/Sharing		25%	12%		
Voting/Voice					
Boss fights	12%				
Competition					
Milestone unlocks					37%
Miniquests		12%		13%	
To attend an appointment	50%	13%			
To win a reward					33%
Build from scratch				13%	
Challenges	12%	12%			
Exchangeable points		57%			
Group quests		100%			
To enhance the identity	12%		38%		

**Table 7 sensors-21-02556-t007:** Selected motives for each mechanic.

	Mechanics	Motives
1	Sharing knowledge	Cooperation
2	Exploration	Exploration
3	Gifting/Sharing	Contribution
4	Voting/Voice	Power/Contribution
5	Boss fights	Power
6	Competition	Power
7	Milestone unlocks	Achievement
8	Miniquests	Exploration
9	To attend an appointment	Affiliation
10	To win a reward	Achievement
11	Build from scratch	Achievement
12	Challenges	Achievement
13	Exchangeable points	Cooperation
14	Group quests	Cooperation
15	To enhance the identity	Autonomy/Hedonism

**Table 8 sensors-21-02556-t008:** Mechanics related to the different motivators.

**Mechanics for Cooperation**	**Motives**
Sharing knowledge	Cooperation
Exchangeable points	Cooperation
Group quests	Cooperation
**Mechanics for Exploration**	**Motives**
Exploration	Exploration
Miniquests	Exploration
**Mechanics for Contribution**	**Motives**
Gifting/Sharing	Contribution
Voting/Voice	Power/Contribution
**Mechanics for Power**	**Motives**
Voting/Voice	Power/Contribution
Boss fights	Power
Competition	Power
**Mechanics for Achievement**	**Motives**
Milestone unlocks	Achievement
To win a reward	Achievement
Build from scratch	Achievement
Challenges	Achievement
**Mechanics for Autonomy/Hedonism**	**Motives**
To enhance the identity	Autonomy/Hedonism
**Mechanics for Affiliation**	**Motives**
To attend an appointment	Affiliation

## Data Availability

Available online https://lasalleuniversities.sharepoint.com/:f:/r/sites/doa-ddr-gretel/Shared%20Documents/2021-Publications/Sensors-Eva-DataMotivators?csf=1&web=1&e=zs0qFF (accessed on 16 March 2021).

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
