# Peer review of "Qualitative Assessment of Effective Gamification Design Processes Using Motivators to Identify Game Mechanics"

_sensors, 2021, doi:10.3390/s21072556_

Round 1

Reviewer 1 Report

General consideration

Main issues:

  • The paper needs an extensive re-writing activity. Some parts are really not easy to follow, making it very difficult to understand the paper.
  • At the beginning of the paper, the authors talk about “Human Centred”, then section 2 is titled “Human Centered ..” and in section 2.2, they talk about User Centred Design. This is not only confusing considering the terminology used, but it raises questions about the authors' knowledge about this approach. In fact, the main goal of the UCD is to include the users in the design process, considering their needs and requirements. In this paper, the authors talk about “expert in the sector since they have knowledge of the applicability of the system”. (and later, “expert consultants in gamification” and then “usability experts” – three different definitions..). Experts are not the users. In fact, at the end of the paper, the authors, discussing the Game4City 3.0 project, talk about students, and the beginning they wrote “Game4City is designed to facilitate 58 the participation of non-specialized users in the visualization – non-specialized users”.
  • It is not clear how user experience (nominated across the paper) is analyzed or took into consideration.
  • The introduction is very confusing, and, in the end, the reader has not a clear idea about the objective of the study.
  • The study has been performed in a defined context (that clearly impacts the results).
  • Why the authors talk about “Qualitative assessment”? they performed a questionnaire and analyzed the data in a qualitative way (the results are clearly numerical). The sentence “For the development of our objectives we have opted for a qualitative approach, since we are in front of an exploratory process with the participation of usability experts.” Doesn’t make any sense. (and why usability experts?)
  • The methodology used is not clear at all, too many details are missing.
  • Interesting Related works are missing.
  • ….

To conclude, I would like to stress that I do think the paper could present interesting results but the content in which the study is framed is completely messy and it makes it impossible to understand the relevance of the presented results.

I really recommend the authors improve and refine the paper and submit it to another journal (it is not really clear to me why they selected “Sensors”). Results, presented in an adequate way, can be interesting for the related community.

Some other detailed comments (the list is not exhaustive)

Abstract

The abstract is not really clear.

What is the meaning of this sentence?

“By doing this approach, during the designing phase of a gamification system, the necessary information needed to obtain a more accurate approximation of user’s profiles who will eventually become players is available, as well as to adapt any gamified proposal in function of the device or sensor to be implemented.”

Also this sentence “[…] which allows the user to organise, classify and decide.” (what?)

Please, remove the possessive in “system’s design”

In general, I strongly recommend the authors rewrite it. In the current version, it is not very clear and informative.

Keywords

“Strategy” doesn’t seem an appropriate keyword

Introduction

This sentence doesn’t make any sense: “Both gamification and user experience are Human Centred [1] systems“ Please, check better the content of the cited paper.

In: “Gamification of any content, system, or environment requires the previous studies in order to stablish the relationship with devices, sensors, systems and methods, in on the other hand the end-users”.  What does mean “requires the previous studies”? and please, check the English (“in on the”).

In general, I find the first paragraphs very confusing.

Here: “Emotional behaviour is based on the emotions that happen to users during the execution of the test”.. which test?

“And the state of mind or humour in which he is” not he but s/he

“a gamified system, sensor, or device” how is it possible to gamify a sensor or a device??

“a gamified environment depending on the sensor or sensors used: fixed, mobile, multi-user environments, etc.” why sensors??

“When talking about the feelings pre-associated by the user, one works from the point of view of intrinsic factors, such as motivation” not clear

These previous experiences make me associate feelings with the product. – what?

Which is the meaning to have figure 1 that it is not even cited in the text content?

Section 2

Why does section 2.1 start with an example on “calibration game”?

“The user experience as a methodology, allows to qualitatively assess the  perspective of the expert profiles in the disciplines being worked on and who are really 238 the future users of the study results” user experience as a methodology??

Conclusion

The first sentences seem to present a different study. This study is about the engagement of 14 experts in answering a simple questionnaire, no  User Centred Design (UCD) system, no qualitative data (no interviews).

Author Response

Dear Reviewer,

Thank you so much for your time, suggestions, and effort to improve our paper. We have answered all your questions and comments, as well as the other reviewer's suggestions improving our contribution (see files). Also, we have done a Proof Editing Review of the document in order to ensure the clarity of the explanation, and avoid any type of orthographical or grammatical error. We hope that this new version will be acceptable for you.

Best.

Reviewer 2 Report

The work concerns the interesting problem of the choice of motivators and gamification mechanisms. It concerns the design of the Game4City game, which is to benefit from the conclusions of this work. However, I have doubts about both the edition of the paper and, in my opinion more importantly, incorrect or insufficiently described research assumptions. The presented case is too general to be referred to.

1. In chapter 2 you present the assumptions about the conducted research. In particular, in chapter 2.2 you first describe (lines 227-232) the User Centered Design process, where the end user of the product is at the center of the design process, and then (lines 260-264) you write that you are testing 14 expert consultants with 5 years of experience. It would be beneficial for the article to clarify this.

2. Source [51] ("Why You Only Need to Test with 5 Users.") does not correspond with the description of the research method because the source deals with testing on only 5 users, while you write that you are testing on 14 consultants.

3. It would be beneficial for evaluation if the text of the questionnaire was presented, it would complement the description of the research method.

4. The content of table 2 is a repetition of the text from lines 202-221.

5. The content of Table 4 is an unnecessary repetition of the highlighted excerpt from Table 3.

6. I propose to move table 5 and description from line 330-338 to the end of chapter 3.2.

7. The description on lines 346-383 repeats what is shown in fig. 2. I suggest that one of them be dropped.

8. Line 474: It looks like the sentence is incomplete.

9. Fragment in lines 545-549: "The game mechanics with a higher degree of dispersion is that of Build from scratch since it is divided between 5 different motives" - why do you think so? - this seems to me to be an unreasonable statement.

10. Fragment from line 552 "... ensure a more effective gamification.". Again unjustified statement - you surveyed 14 people, why do you claim that gamification is more effective?

The work should be supplemented with an analysis of the use of the designed game from the point of view of its users. Only then can it be stated whether the applied gamification mechanisms are effective.

Author Response

(The authors gave the same response as above.)

Reviewer 3 Report

The paper presents an interesting insight into the opinions of expert consulters in gamification in order to establish relations between motivators and game mechanics, which can lead to designing more effective gamified systems. The paper is well written, methodology is sound and the results are well presented. The only thing that I would add to the research is the inclusion of more qualitive data and comments from the experts, since their knowledge is the most valuable part of the study. Maybe a supplement with all the comments regarding different mechanics and motivations could be very useful to get more insight into expert opinions.

Author Response

(The authors gave the same response as above.)

Reviewer 4 Report

The study aims to establish relations between motivators with game mechanics. The results of the qualitative assessment with expert consulters in gamification are presented.

Comments:

  1. The study should have the research questions formulated, which are answered by the results of the survey.
  2. Motivators depend on psychological types of players such as the ones established by the HEXAD questionnaire. Discuss this aspect of your research.
  3. Line 74: confusing statement. What declines? A player or a commitment? Rewrite for clarity.
  4. The study should also discuss the Computational Evaluation of Effects of Motivation Reinforcement on Player Retention.
  5. Line 266: “Conformity of respondents”. Should it be “Consent of respondents”?
  6. Table 6, Table 7: percentages should sum to 100%. However, the values presented do not obtain 100% both in rows and in columns. So what is the meaning of these values?
  7. The results of this study should be linked with and compared against other studies. However, I do not see any comparison with related works in this study.
  8. Line 581-583: rather an unsubstantiated claim that is not supported by this study. I suggest to remove it.
  9. The game mechanics should be explained in better detail and visualized, if possible. You can describe game mechanics using the Machinations diagrams https://machinations.io/
  10. The statistical reliability of the survey results must be evaluated.

Author Response

(The authors gave the same response as above.)

Round 2

Reviewer 1 Report

This version is a lot better in terms of readability and clarity.

Unfortunately, I still think there are different points that need to be better addressed by the authors. Reading the introduction and the first sections, the reader has a high expectation about the study, but then, the results are framed in a very specific context: engaging 14 gamification experts (consultants) in answering a simple questionnaire. I suggest the authors better frame the defined context from the beginning of the paper (removing all the extra not-needed contextual information). I decided for "major revision" recommendation, but I highly recommend the authors refining the paper (using all the needed time) and send it to a more appropriate venue.

Again, I think this paper has potential and findings could be interesting to the right community who could really benefit from the results, but, right now, it still needs to be improved, refined to better frame the actual context of the study.

To report just some issues at the very beginning of the paper (several others are present in all the document content).

I suggest changing the sentence “Both gamification and  user  experience are human-centred systems” because a system can be gamified without considering the user’s need, the same issue occurs with user experience that is something that a system can or can not reach. A system can be designed using a human-centred approach, but it could still not able to reach a good user experience.

This sentence “The gamification of any content, system or environment requires prior studies to understand the technology and technique of the use of devices, sensors, systems or methods” is very generic and seems just a way to justify the article in the special issue journal.

This sentence: “Starting from these premises, players and users become protagonists of the development process, promoting the creation of more effective systems from the beginning”  I completely agree, but then, the paper focuses on “expert gamification consultants”.

Author Response

Dear Reviewer,

Thank you so much for your review and comments. We have clarified better our statements and ideas following your suggestions in the new version of the paper.

Best Regards.

Reviewer 2 Report

The authors referred to my comments. I think the quality, soundness of the paper has improved. Thus, I suggest accepting the paper.

Author Response

(The authors gave the same response as above.)

Reviewer 4 Report

Well revised. I have no further comments.

Author Response

Dear Reviewer,

Thank you so much for your review and comments. We have clarified better our statements and ideas following your suggestions in the new version of the paper.

Best Regards.

This manuscript is a resubmission of an earlier submission. The following is a list of the peer review reports and author responses from that submission.

Round 1

Reviewer 1 Report

The paper is very interesting for gamification experts.  The research presents a study with expert consulters in gamification to measure the relations between motivators and game mechanics. 

The paper is well described and easy to read. The main drawback is the number of experts involved in the study; are the authors confident that 14 subjects is sufficient to state general opinions?

Also, there are too many articles from the same authors in the bibliography, please review the articles using other researchers' work as well.

Reviewer 2 Report

The paper addresses a research topic related with the relationship between motivators and games mechanics based on a survey of expert consultants.

The sample size is small (which is understandable, since the sample is made up of expert consultants who are a very specific professional profile), and the statistical study performed is not very thorough (based only in relative frequencies) and with a lot of redundant information that needs to be corrected and restructured. The study conducted could have been reinforced with qualitative research tools that would have allowed more robust conclusions to be drawn, such as, for example, an interview with the experts. Even the questionnaire could have been designed using a Likert scale that would have allowed more quality information to be extracted.

Likewise, I do not see a direct relationship between the subject matter of the article and the scope of the journal. Therefore, I cannot recommend it for journal publication in Sensors although it may be of interest in journals more related to the subject.

In addition, the following aspects should be corrected or improved:

“Mechanics” is not a keynote related with the contribution of the paper.

Table 2 is redundant because the reasons are numbered in the text.

Authors indicate “14 expert consultants in gamification [38] with more than 5 years of experience in business projects, educational projects or research projects, and with a standard deviation of 3.33 with respect to age.” Since the deviation is indicated, the mean must be also mentioned. Define better the sample. How has been the 14 experts selected? In addition, the profile of gamification consultants can be very diverse and research results can vary accordingly. It is advisable to include as an annex an anonymized description of the consultants' background: type of companies they have worked for, field of application of gamification, size of the projects, if academy or professional profile, etc.

Authors indicate “the research has followed the model indicated by Universitat Ramon Llull, both for the information detailing the research, and the type of informed consent given to the participants.” “Model” is maybe not appropriate, perhaps the term is ethical standards or ethical regulations.

In 2.2.3. sections authors only indicate that the questionnaire is telematic. Explain better the details of the process, the order of the questions, the configuration, the realization data, etc.

Table 5 is confusing. It tells about the things that are not considered games mechanical and each one is evaluated with 0 %. Does this mean that they are considered or not considered at all? It is confusing for the reader. Maybe this table is redundant because it is already explained in the text.

The pie charts shown from Fig 3 to Fig 11. are not suitable to visualize so many items, they overload the article and impair the overall view of the results. Furthermore, these data are included in Table 7 and 8, so the diagrams do not give additional information. Please remove them or look for another way to show the data in a more compact form.

In Table 9 the selected motives shown are the ones with have a higher relative frequency. This must be better explained in the text or in the footer.

Limitations of the research due to the sample size and the lack of other research tools should be indicated in the discussion. Even so, the research must be improved in its formal aspect and in its content.

Reviewer 3 Report

OVERVIEW

This study wants to analyze the relationship between game mechanics and motivators. A telematic survey is sent to 14 highly qualified experts. The research begins with the selection of game mechanics between those determined by the Card Sorting analysis. Then a series of motivators are chosen following the Wheel of motives. Finally, the experts linked the series of game mechanisms with motivators to define the user profile for the decision in the game design.

ASSESSMENT

I think the topic is interesting and the potential is high. However, this study seems more a pilot study rather than a fully-fledged one. 

Also, I have specific serious concerns:

  • I would not say “experiment” or speak anywhere of causal effects. Data  were collected from a simple questionnaire, just confronting the percentage of the respondents. I would say “survey” instead the “experiment”. Moreover, the sample seems really small to claim that those game mechanisms and motivators can be applied for everyone. Statistically, nothing can be actually said on effects or causation, so that no claim in this regard should be made.

  • The description by means of percentages is fine, but I would like to see more: can you ask experts to motivate, with their words, why they linked that particular game mechanism with that motivator?

  • The description of results is too long and redundant: in the text authors can describe the results shown in the tables or in the pie charts (combining the graphs in only one or, at most, two). I would include only one between tables and charts in the appendix, because is useless to have both in the main text given that both report same results and numbers.

  • I think that something is missing in the paper: as the authors claim at the end, it could be interesting to use those results in some project and analyze it.

  • Finally, I found some typos in the text and, especially, in figures (i.e., “Afiliation” in all pie charts)